# Scoping review protocol to map evidence on South-South learning exchange in family planning

Komal Preet Allagh ![ORCID],[1] James Kiarie,[2] Isotta Triulzi,[3] Rita Kabra ![ORCID] [2]

[1]Consultant-Department of Reproductive Health and Research, World Health Organization, Geneva, Switzerland
[2]Department of Reproductive Health and Research including UNDP/UNFPA/UNICEF/WHO/World Bank Special programme of Research, Development and Research Training in Human Reproduction, World Health Organization, Geneva, Switzerland
[3]Institute of Managment, Scuola Superiore Sant'Anna, Pisa, Italy

**Correspondence to**
Dr Rita Kabra; kabrar@who.int

## ABSTRACT

**Introduction** South-South learning exchange (SSLE) is an interactive learning process where stakeholder teams exchange knowledge and experience to help one, or both to work towards change, by identifying, adopting and/or strengthening implementation of a best practice. SSLE has been conducted between countries to share knowledge on best practices and policies in family planning. To the best of our knowledge, no scoping review has been conducted to synthesise evidence on SSLE in family planning. In this paper, we outline the protocol to conduct scoping review on SSLE in family planning.

**Methods and analysis** Arksey and O'Malley's scoping review framework with adaptions from Levac *et al* will be used to guide this scoping review. We will search electronic databases (Medline, Embase, CINAHL, Hinari, ProQuest DB, PUBMED, Web of Science and WorldCat), grey literature sources and reference lists of included studies. We will focus on literature published till August 2022. The abstract and title screening, full-text screening and data charting will be conducted by two independent reviewers. The findings will be summarised into a narrative based on thematic analysis. Stakeholder interviews will be conducted to understand their perception and experiences in applying SSLE in family planning.

**Ethics and dissemination** The ethics review committee at WHO, Geneva, has exempted this study from ethical approval (ERC.0003752). The findings from the study will provide useful insights into effective approaches, barriers, facilitators to conduct SSLE in family planning. This knowledge will be of significant public health relevance and will help in designing future learning exchanges between countries in the south to accelerate access to quality family planning services. The findings will be disseminated via peer-reviewed journals, conference proceedings, newsletters and workshops.

## STRENGTHS AND LIMITATIONS OF THIS STUDY

⇒ This will be the first scoping review to identify effective approaches, barriers, facilitators to conduct South-South Learning exchange in family planning.
⇒ Stakeholders will be consulted and engaged throughout the review process.
⇒ A comprehensive search strategy is developed with the consultation of a chief librarian to promote a sensitive search.
⇒ The quality of the included studies will not be assessed.

or local administrative units. The SSLE can offer value to both teams by supporting bidirectional sharing of knowledge, good practices, supporting scaling-up of good practices and capacity building of participants to advocate for the change process. SSLE is often far more convincing and contextually appropriate than learning from publications or experts.[2] Learning exchanges have focused on a wide range of topics in the past, from trade, finance, food security, nutrition and health.[3]

Since 2019, WHO has been conducting SSLE in family planning under the WHO Family Planning Accelerator project[4] using a standardised five-step methodology[1] designed to ensure that SSLE is country driven, focused on outputs/outcomes and involves rigorous monitoring of the process. The WHO FP Accelerator project builds on the WHO FP Umbrella project that supported over 50 countries to update their national Family Planning (FP) guidelines to the latest WHO recommendations.[5]

At the International Conference on Population and Development, Cairo in 1994, one of the recommendations was that more attention should be given to South-South cooperation (SSC) as an important instrument of development.[6] Since then, several countries have reported using SSLE referred to as SSC, to improve family planning outcomes using different approaches like study tours, site

## BACKGROUND

The WHO describes South-South learning exchange (SSLE) as an interactive learning process where stakeholder teams exchange knowledge and experience to help one or both teams to work towards change by identifying, adopting and/or strengthening implementation of a best practice.[1] Learning exchange can take place between two teams at the country level or within countries, between provinces, regions, states, districts

visits, training or expert visits. Together with political dialogue, technical and financial cooperation, SSLE has assisted in several knowledge and expertise exchanges through programmes, projects and initiatives that have helped solve specific challenges. However, to date, no document highlights and summarises the key features of an SSLE in family planning, such as the most effective approach to achieve the FP outcomes, the barriers and facilitators to expect from the learning exchange and how to overcome these, and the objectives of the SSLEs that have already taken place.

To the best of our knowledge, there is no existing published synthesis on SSLE in family planning that incorporates both the reviewed and non-peer-reviewed literature. This scoping review aims to systematically review published and grey literature on SSLE in family planning and identify gaps in the available knowledge to help guide future SSLEs in family planning and research in the field. The study objectives are to review published and grey literature on SSLE in family planning to identify the (1) purposes, (2) approaches, (3) key outcomes and (4) enablers and barriers.

## METHODS

This scoping review is based on the framework proposed by Arksey and O'Malley's,[7] which has been further developed by Levac et al.[8] The framework has six steps: (1) identification of the research question, (2) identification of relevant studies, (3) study selection, (4) data charting, (5) collating, summarising and reporting the results and (6) stakeholder consultation

### Stage 1: identification of the research question

This scoping review seeks to answer the following specific research questions:

1. For what purposes have South-South learning exchanges in family planning been used?
2. What approaches and methods are used in conducting South-South learning exchange in family planning between countries?
3. What are the barriers and facilitators encountered in conducting South-South learning exchange in family planning?
4. What outcomes have been achieved by South-South learning exchange in family planning?

### Stage 2: identification of relevant studies (search strategy)

With support from the chief librarian at WHO Geneva, the authors will conduct a systematic search within the following electronic databases: Medline, Embase, CINAHL, Hinari, ProQuest DB, PUBMED, Web of Science and WorldCat. This study will use a complete search strategy (online supplemental file 1) that employs keywords, medical subject headings (MeSH) or subject headings search terms that relate to key review concepts: South-South learning exchange, South-South knowledge exchange, South-South exchange, Peer to peer learning exchange, SSC, information sharing, information exchange, knowledge sharing, knowledge exchange, learning exchange, Family planning, contraception, reproductive health, as well as Boolean terms "AND" and "OR".

No language restrictions will be imposed. Reference lists of the articles will be used to identify more studies using a snowball approach. We will also conduct a grey literature search using Google Scholar and relevant websites (FP 2030, Partners in population and development, UNFPA, WHO, UNOSCC, S-S Galaxy, World Bank and USAID). Types of grey literature that will be retrieved and used in the review include reports, articles, conference proceedings, PhD and Master's thesis, and case studies.

### Stage 3: study selection

This review will include studies that meet the defined eligibility criteria (see table 1), published till August 2022. No limits will be placed on study period, language and location of study.

### Selection process

We will use a two-part study selection process. First, two independent reviewers (RK and KPA) will screen all

| Table 1 | Eligibility criteria for the scoping review | |
|---|---|---|
| **Criteria** | **Inclusion** | **Exclusion** |
| Country | Any country | None |
| Date | Any year | None |
| Language | All languages | None |
| Research focus | Studies that mention objectives, purpose, approaches, process, enablers, barriers and outcomes of SSLE in family planning. | SSLE on topics other than Family Planning |
| Geographical location | SSLE between countries | SSLE within a country (between institutions, cities and districts) |
| Document type | Scientific report, case study, commentaries, research article, conference proceedings, student thesis, letter to editors and reviews. | Newspaper, power point presentations and magazine articles |
| SSLE, South-South learning exchange. | | |

**Table 2** Data extraction framework

| Main category | Description |
|---|---|
| 1. Authors | Name of the authors |
| 2. Title | Title of study |
| 3. Source of publication | Title of the journal or name of publishing organisation |
| 4. Year of publication | Year of publication |
| 5. Purpose/objective of SSLE | Describe the stated purpose of the SSLE |
| 6. Year and duration of the SSLE | Indicate the year of SSLE and duration the SSLE lasted |
| 7. Countries participating in the SSLE | List the mentor and mentee countries. |
| 8. Approach used for SSLE | Describe the method used to conduct the learning exchange, example: study tour, virtual, reciprocal exchange, etc. |
| 9. Process of SSLE | Describe the stated process of the SSLE for instance: was a standard method used, planning phase, were stakeholders involved, was an action plan developed, was it implemented, was a follow-up done after SSLE was completed. |
| 10. Key stakeholders | Indicate the key Family Planning stakeholders involved in the SSLE in both the countries |
| 11. Reported outputs | Describe the family planning outputs reported in the SSLE (eg, capacity building, policy change, etc.) |
| 12. Reported outcomes | Describe the family planning outcomes reported in the SSLE (eg, contraceptive prevalence rate, unmet need) |
| 13. Barriers | Describe the factors that inhibit the implementation of the SSLE |
| 14. Facilitators | Describe the factors that supported or enabled the implementation of the SSLE |

SSLE, South-South learning exchange.

titles and abstracts as per inclusion/exclusion criteria. Any article that is considered relevant by either or both reviewers will be included for full-text review. The reviewers (RK and KPA) will then independently assess the full text to determine if the study is included. A third reviewer (JK) will resolve any discrepancies that arise between the two reviewers. Zotero reference manager V.5.0.96.3,[9] a citation and bibliographic software will be used to store and organise all references. The Preferred Reporting Items for Systematic Reviews and Meta-Analyses (PRISMA) flow diagram[10] will be used to report the results of the screening.

### Stage 4: data charting
A data charting form will be prepared in Microsoft Excel to extract study characteristics. KPA and RK will extract all relevant data from the included studies after a thorough reading of the full texts. The data charting form will include 14 categories as listed in table 2. This list will be updated when we start reviewing the studies to capture all relevant data to answer the review questions.

The data charting form will be pilot tested in a few studies to ensure it is capturing all the information accurately.

### Stage 5: collating, summarising and reporting the results
A narrative account of the included studies will be prepared to present the literature on SSLE in family planning based on a thematic analysis. Each author will independently review the extracted information to summarise the findings presented across the articles. Relevant themes and subthemes relating to the study objectives will be developed around the following: (1) purpose of SSLE in family planning, (2) methods and process used for SSLE in family planning, (3) FP outputs and outcomes achieved from SSLE in family planning and (4) barriers and enablers of SSLE in family planning.

### Stage 6: stakeholder consultation
Levac *et al* suggested that the consultation stage provides opportunities for stakeholder involvement, providing insights beyond what is reported in the literature.[8] The consultations will aim to understand stakeholder perception and experiences in applying SSLE in family planning. Following ethical approval, one reviewer (IT) will identify and invite the authors of the included studies for the interviews. In addition, a snowball approach will be used to identify more experts. We will interview 12–20 individuals. All interviews will be conducted virtually via google meet for an approximate duration of 45–60 min and will be recorded using a voice recorder and through taking notes. Participants will be assured of confidentiality and verbal informed consent will be recorded. Each interview will be transcribed verbatim. Relevant themes and subthemes relating to the study objectives will be identified using RQDA software for qualitative analysis.[11]

The interviewer will provide an overview of the scoping review. A semistructured interview guide is developed to understand the experiences and views of stakeholders on SSLE in family planning. The questionnaire includes a set of open-ended questions (online supplemental file 2) to guide the discussion and will cover the following topics: (1) role of the organisation in SSLE and the process used (2) purpose of SSLE and approaches used (3) views and experiences on SSLE in FP (4) perception on challenges and successes observed during the SSLE process and (5) lessons learnt.

### Patient and public involvement

No patients are involved. The study is a review of literature, there are no study participants.

## DISCUSSION

The proposed review aims to map evidence on SSLE in family planning. The extracted data will be presented in a narrative and tabular form to cover all the review objectives. The results of this scoping review will build evidence on effective approaches, barriers, facilitators and key outcomes achieved in SSLE in family planning and may reveal further research areas. Results of the scoping review will be disseminated in a peer-reviewed journal and other ways such as through consultations, newsletters, conferences and workshops.

## ETHICS AND DISSEMINATION

The ethics review committee at WHO, Geneva, has exempted this study from ethical approval (ERC.0003752). The findings from the study will provide useful insights into effective approaches, barriers, facilitators to conduct SSLE in family planning. This knowledge will be of significant public health relevance and will help in designing future learning exchanges between countries in the south to accelerate access to quality family planning services. The findings will be disseminated via peer-reviewed journals, conference proceedings, newsletters and workshops.

**Contributors** KPA prepared the first draft with significant contributions from JK, IT and RK. All authors reviewed the draft manuscript and approved the final manuscript for publication.

**Funding** This work received funding from the WHO through the WHO FP Accelerator Project 2019-2022 supported by the Bill & Melinda Gates Foundation (OPP1203035).

**Competing interests** None declared.

**Patient and public involvement** Patients and/or the public were not involved in the design, or conduct, or reporting, or dissemination plans of this research.

**Patient consent for publication** Not applicable.

**Provenance and peer review** Not commissioned; externally peer reviewed.

**ORCID iDs**
Komal Preet Allagh http://orcid.org/0000-0002-6272-2135
Rita Kabra http://orcid.org/0000-0001-6595-2035

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
