## [Reviewer comments · BMJ Open]

ARTICLE DETAILS

TITLE (PROVISIONAL)	A scoping review protocol to map evidence on South-South learning exchange in family planning
AUTHORS	Allagh, Komal Preet; Kiarie, James; Triulzi, Isotta; Kabra, Rita

VERSION 1 – REVIEW

REVIEWER	Jiang, Hong Fudan University, School of Public Health
REVIEW RETURNED	12-Sep-2022

GENERAL COMMENTS	This protocol aims to conduct a scoping review on South-South learning exchange in family planning based on an adapted existing framework. Overall, the protocol is well written. The research questions are meaningful. The searching strategy is comprehensive. The stakeholder consultation will help to interpret and provide additional information for South-South learning exchange in family planning. The study will help to share knowledge and promote evidence-based practices in family planning for different countries and regions. There are only a few minor comments. 1. It seems there are duplications of definitions in South-South learning exchange in the first paragraph and by WHO in the third paragraph of Background. These two definitions are very similar.2. When searching for literatures, is there restriction in countries where the studies were conducted?
---

REVIEWER	Pamungkasari, Eti Universitas Sebelas Maret, Public Health
REVIEW RETURNED	08-Dec-2022

GENERAL COMMENTS	The discussion section still needed to explore. For example author can explain probable limitation and strength of review findings. References used in this protocols manuscript are still limited.
---

REVIEWER	Neiterman, Elena University of Waterloo Faculty of Applied Health Sciences
REVIEW RETURNED	22-Jan-2023

GENERAL COMMENTS	Thank you for the opportunity to review this protocol for a scoping review on South-South learning exchange in family planning. The authors' methodology relies on the framework of Arksey and O'Malley's with additional insights from Levac et al. The protocol is written well and provides step by step outline of the proposed research activities.
--

	There are only a few additional suggestions that I have for the authors:  1. The authors propose to include a search of eight different databases. While the search strategy seems comprehensive, I am wondering if the authors can clarify which databases will be used (primarily) for the search of grey literature and which ones for academic literature. I also noticed that most databases that are planned to be searched are health related, and it would be good for the authors to identify which databases would yield papers that more connected to social sciences (e.g. such as Scopus). 2. It is my understanding that the authors plan to include into their review not only empirical studies, but also various reports, papers and commentaries that are not empirical. Still, I am wondering if for the empirical studies the authors would want to report on the methods and thus have this information included in their charting document and findings section. 3. Relying on the additions of Levac et al to the scoping review methodology of Arksey and O'Malley, the authors propose to include an extensive consultation with stakeholders, whom they identify as the authors of the collected papers. I would propose to provide more clarifications for the need/rationale for this step. Usually, interviews with stakeholders are sought to identify what the literature search missed, what topics are not covered in the existing literature, or what would be fruitful avenues for knowledge translation. However, the authors propose to explore the experiences and views of stakeholders on SSLE in family planning, a topic that they might have already covered in their paper and/or would not shed more light on the results of the scoping review. Moreover, the interview guide that the authors proposed to use does not explicitly link the interviews to the results of the scoping review, which makes this step seem like as a stand-alone study on people's experiences and views of SSLE in family planning, and not a consultation stage for the scoping review. To address this, I would propose that the authors consider if this last stage would constitute a research project on its one (e.g. a qualitative study on the experiences and views of SSLE in family planning) or provide a stronger rationale and a link to the scoping review methodology they outlined so well. I hope these comments can help the authors and wish them all the best with their research.
--	--

VERSION 1 – AUTHOR RESPONSE

Reviewer: 1

Prof. Hong Jiang, Fudan University

Comments to the Author:

This protocol aims to conduct a scoping review on South-South learning exchange in family planning based on an adapted existing framework. Overall, the protocol is well written. The research questions are meaningful. The searching strategy is comprehensive. The stakeholder consultation will help to interpret and provide additional information for South-South learning exchange in family planning. The study will help to share knowledge and promote evidence-based practices in family planning for different countries and regions.

Response: Thank you

There are only a few minor comments.

1. It seems there are duplications of definitions in South-South learning exchange in the first paragraph and by WHO in the third paragraph of Background. These two definitions are very similar.

Response: Thank you for pointing out the duplication in sentences. We have deleted the second sentence on the South-South learning exchange definition in the third paragraph. The first sentence is modified to 'The World Health Organization (WHO) describes South-South learning exchange (SSLE) as an interactive learning process where stakeholder teams exchange knowledge and experience to help one or both teams to work towards change by identifying, adopting and/or strengthening implementation of a best practice'

2. When searching for literatures, is there restriction in countries where the studies were conducted?

Response: No, there were no limits placed on the location of the study, as mentioned in the methodology under step 3 (study selection) on page 6.

Reviewer: 2

Dr. Eti Pamungkasari, Universitas Sebelas Maret

Comments to the Author:

The discussion section still needed to explore. For example, author can explain probable limitation and strength of review findings. References used in this protocols manuscript are still limited.

Response: Thank you for your suggestions. The strengths and limitations of the study have already been included at the beginning of the manuscript (right after the abstract) and hence were not included in the discussion section.

We have added a few more references in the introduction section.

Reviewer: 3

Dr. Elena Neiterman, University of Waterloo Faculty of Applied Health Sciences

Comments to the Author:

Thank you for the opportunity to review this protocol for a scoping review on South-South learning exchange in family planning. The authors' methodology relies on the framework of Arksey and O'Malley's with additional insights from Levac et al. The protocol is written well and provides step by step outline of the proposed research activities.

Response: Thank you

There are only a few additional suggestions that I have for the authors:

1. The authors propose to include a search of eight different databases. While the search strategy seems comprehensive, I am wondering if the authors can clarify which databases will be used (primarily) for the search of grey literature and which ones for academic literature. I also noticed that most databases that are planned to be searched are health related, and it would be good for the authors to identify which databases would yield papers that more connected to social sciences (e.g. such as Scopus).

Response: Thank you for your feedback. We developed the search strategy for this review in consultation with the chief librarian. In addition to the searches on the electronic databases mentioned in the manuscript, we will search the following websites for grey literature for reports and papers on south-south learning exchange in family planning: FP 2030, Partners in population and development (PPD), UNFPA (United Nations Population Fund), WHO, UNOSCC (United Nations Office of South-South cooperation), South-South Galaxy, World Bank, UNDP (United Nations Development Programme) and USAID (United States Agency for International Development). This information has already been included in the manuscript on page 6, under step 2 of the methodology section.

2. It is my understanding that the authors plan to include into their review not only empirical studies, but also various reports, papers and commentaries that are not empirical. Still, I am wondering if for the empirical studies the authors would want to report on the methods and thus have this information included in their charting document and findings section.

Response: Thank you for the suggestion. In table 2 on the data extraction framework (page 8), point 8 is on the approach used for South-South learning exchange, which is the method/approach the countries have used for the learning exchange.

3. Relying on the additions of Levac et al to the scoping review methodology of Arksey and O'Malley, the authors propose to include an extensive consultation with stakeholders, whom they identify as the authors of the collected papers. I would propose to provide more clarifications for the need/rationale for this step. Usually, interviews with stakeholders are sought to identify what the literature search missed, what topics are not covered in the existing literature, or what would be fruitful avenues for knowledge translation. However, the authors propose to explore the experiences and views of stakeholders on SSLE in family planning, a topic that they might have already covered in their paper and/or would not shed more light on the results of the scoping review. Moreover, the interview guide that the authors proposed to use does not explicitly link the interviews to the results of the scoping review, which makes this step seem like as a stand-alone study on people's experiences and views of SSLE in family planning, and not a consultation stage for the scoping review. To address this, I would propose that the authors consider if this last stage would constitute a research project on its one (e.g. a qualitative study on the experiences and views of SSLE in family planning) or provide a stronger rationale and a link to the scoping review methodology they outlined so well.

Response: Thank you for your comment. We have contacted the authors of the included studies as a starting point to identify experts in South-South learning exchange in family planning. We have identified and interviewed other experts in the area through these authors (snowballing technique). Hence, the interviews will help gather information beyond what is mentioned in the included studies. The interviews were semi-structured, and the questions in the guide were all developed around the objectives of the scoping review.

We are also considering publishing the results of the stakeholder interviews separately as a qualitative study, as suggested by you also.

I hope these comments can help the authors and wish them all the best with their research.

VERSION 2 – REVIEW

REVIEWER	Neiterman, Elena University of Waterloo Faculty of Applied Health Sciences
REVIEW RETURNED	10-Feb-2023
GENERAL COMMENTS	The authors addressed all the comments. I have no additional suggestions and wish them best of luck with the study.